# Detection versus Instance Segmentation for Multi-Species Malaria Diagnosis: A Head-to-Head Comparison and Multi-Dataset Validation of YOLOv12 Architectures with Small Object Optimization

**Ahmed Tahiru Issah**                                   AISSAH@ANDREW.CMU.EDU
**Idaya Seidu**                                          ISEIDU@ANDREW.CMU.EDU
**Carine Mukamakuza***                                  CMUKAMAK@ANDREW.CMU.EDU
*Carnegie Mellon University Africa*

**Editors:** Accepted for publication at MIDL 2026

## Abstract

Automated malaria parasite detection using deep learning holds promise for addressing diagnostic gaps in resource-limited settings, yet most studies rely on single-dataset evaluations that fail to capture real-world variability. In this work, we rigorously validate YOLOv12-based architectures for malaria detection across diverse geographic and institutional contexts. We introduce a dual-head architecture combining instance segmentation with a high-resolution P2 detection head to target tiny ring-stage parasites. Our evaluation on a diverse Rwandan thick-smear dataset (2,739 images) and two external datasets from Ghana (Lacuna) and Nigeria (FASTMAL) reveals critical insights into model robustness. While the proposed YOLOv12-Seg-N-P2 model achieves state-of-the-art internal performance (mAP@50 0.888) and significantly improves detection of challenging *P. vivax* (+10.9%) and *P. falciparum* ring forms, external validation exposes severe domain shift, with performance dropping by > 80% on unseen datasets. We further demonstrate that while P2 heads enhance morphological precision on source data, they reduce zero-shot generalization, likely by overfitting to dataset-specific acquisition characteristics. We additionally evaluate white blood cell (WBC)-anchored stain normalization and pixel-scale rescaling as inference-time domain adaptation strategies. While WBC detection improves substantially (up to +45% on Lacuna), *P. falciparum* detection remains critically low across both external datasets despite partial recovery on FASTMAL, confirming that preprocessing-based adaptation alone is insufficient for reliable cross-site parasite detection.

**Keywords:** Malaria diagnosis, deep learning, YOLOv12, instance segmentation, object detection, computer vision, medical imaging

## 1. Introduction

Malaria remains a major global health threat, causing an estimated 263 million cases and 597,000 deaths in 2023, with the burden concentrated in sub-Saharan Africa ([World Health Organization](), 2024). Effective control relies on rapid, accurate diagnosis to guide treatment and limit drug resistance. Multiple *Plasmodium* species infect humans, including *P. falciparum*, *P. vivax*, *P. malariae*, and *P. ovale* ([CDC](), 2024), making species identification essential for appropriate treatment ([Centers for Disease Control and Prevention](), 2025). Microscopy of Giemsa-stained smears remains the gold standard but is slow, subjective, and

---

* Corresponding author

dependent on skilled microscopists (Centers for Disease Control and Prevention, 2024). In low-resource settings, thick blood smears are a primary diagnostic tool (Centers for Disease Control and Prevention, 2024; Manescu et al., 2020a).

Thick smears concentrate parasites within a small field, enhancing sensitivity for low parasitemia (Koirala et al., 2022) and are therefore standard in endemic regions (Manescu et al., 2020a). Despite this, deep learning research has focused on thin smears (Akpo et al., 2024), which are less complex than thick smears that include debris, platelets, and white blood cells (Nakasi et al., 2025; Koirala et al., 2022). This mismatch between research and clinical practice underscores the need for modality-relevant validation studies.

Deep learning has expanded automated microscopy capabilities. Early CNN classifiers like VGG19 and InceptionV3 labeled cropped cells as parasitized or uninfected (Okoronkwo, 2025), but these approaches do not reflect real diagnostic workflows. Object detection models, particularly the YOLO family, now identify parasites in whole fields and estimate density (Lipsa et al., 2025), evolving from YOLOv3 and YOLOv4 (Lipsa et al., 2025) to YOLOv5 and YOLOv8 (Koirala et al., 2022; Zedda et al., 2025). Task-specific validation shows YOLOv5 can outperform later versions in speed and memory (Lipsa et al., 2025), while variants like YOLO-mp (Koirala et al., 2022) and YOLO-PAM (Zedda et al., 2023) demonstrate detection-based potential.

Debate exists on whether bounding box detection or instance segmentation is better for parasite identification. Segmentation provides pixel-level precision and is widely used in biomedical imaging (Akpo et al., 2024; Abraham, 2019), but malaria parasites are small, so the additional annotation and computational load may offer limited benefits. No direct comparison of high-performing detection and segmentation models on thick smears exists; this study provides one.

Detecting early ring-stage parasites remains difficult due to their small size. Standard feature pyramid networks lose high-resolution detail at deeper layers, reducing detection of small objects. Introducing a P2 detection head on early feature maps preserves fine spatial detail and may improve detection (Koirala et al., 2022). Previous work has highlighted this challenge (Zedda et al., 2025) and explored small-object modules (Zhang et al., 2024). This study evaluates P2 heads in YOLO Para for measurable improvements.

Finally, reliance on single-dataset validation limits robustness. Domain differences from slide preparation, staining, microscope hardware, and geography can reduce model performance by 5-30% when applied elsewhere (Zedda et al., 2025; Okoronkwo, 2025; Nakasi et al., 2025). Robust external validation is essential, and this study tests zero-shot generalization across datasets from Ghana (Lab, 2023; Nakasi et al., 2025) and Nigeria (Manescu et al., 2020a).

These clinical, methodological, and technical gaps motivate the central aim of this paper. The goal is not to introduce a new architecture but to provide a rigorous validation of existing models. The study examines the relative strengths of detection and segmentation for thick smears, evaluates the benefit of P2 heads for small parasite detection, measures generalization across diverse datasets, assesses computational efficiency for resource-limited settings, and analyzes performance on challenging cases such as rare species and low-parasitemia slides. Through this evidence, the study seeks to clarify best practices and support progress toward clinically deployable automated malaria diagnostics.

## 2. Related Work

### 2.1. Manual and Automated Malaria Microscopy

For more than a century, the examination of stained blood smears under a microscope has remained the gold standard for malaria diagnosis because it is affordable, sensitive, and capable of providing parasite density, species, and life-cycle stage information (Lipsa et al., 2025; Wangai et al., 2011; Poostchi et al., 2018). Despite its strengths, manual microscopy depends heavily on the skill and experience of the microscopist, which leads to substantial variability in diagnostic consistency (Koirala et al., 2022). Thick blood smears, although widely used for detection in endemic regions, introduce additional challenges. The lysis of red blood cells can distort or remove parasites during staining and washing, which may lower visible parasite counts (Bejon et al., 2006). The resulting images contain overlapping layers of parasites, white blood cells, platelets, and debris, which makes accurate identification difficult (Poostchi et al., 2018; Zedda et al., 2025; Koirala et al., 2022; Fatima and Farid, 2020). Digitized microscopy and automated analysis offer a way to reduce subjectivity and workload, and deep learning methods have become central to these efforts.

### 2.2. Evolution of Deep Learning Architectures for Object Detection

Deep learning for medical imaging has advanced rapidly, especially in object detection. Early work was dominated by two-stage detectors such as the R-CNN family, which proposed candidate regions before classification. These models achieved high accuracy but were often too slow for real-time use (Lipsa et al., 2025; Zedda et al., 2025). One-stage detectors, including the YOLO family, removed the region proposal step and predicted bounding boxes and classes directly, which made them faster and more suitable for real-time applications (Lipsa et al., 2025; Koirala et al., 2022). Successive YOLO versions improved this balance between speed and accuracy, although earlier versions sometimes outperform later ones in efficiency (Lipsa et al., 2025). Many modern models also incorporate attention modules such as CBAM, which highlight important features like parasite nuclei and suppress irrelevant background patterns (Zedda et al., 2025; Zhang et al., 2024). These innovations have been essential in adapting general vision models to medical microscopy.

### 2.3. Deep Learning Applications in Malaria Detection

Early deep learning studies treated malaria diagnosis as a simple classification task, using CNNs to label isolated cell patches as parasitized or uninfected (Okoronkwo, 2025). These works showed feasibility but relied on external preprocessing to crop cells, which limited clinical relevance. The use of object detection models marked a major shift. YOLO-based systems can process a full microscopic field, locate parasites, and classify species or stages, which aligns more closely with clinical needs such as estimating parasitemia (Koirala et al., 2022; Zedda et al., 2025). Many studies have refined these detectors and achieved strong results, yet an important methodological gap remains. Object detection offers efficient localization through bounding boxes, while instance segmentation provides more precise pixel-level masks. A direct and rigorous comparison of these two approaches for thick smear parasites is still missing. This study addresses that gap.

## 2.4. Segmentation in Medical Imaging

Semantic segmentation provides pixel-level masks for each object rather than simple bounding boxes (Akpo et al., 2024). This level of detail is important in areas such as oncology or anatomical analysis, where accurate boundaries influence treatment decisions and measurement (Athalye and Arnaout, 2023; Akpo et al., 2024). U-Net and its variants remain foundational architectures for biomedical segmentation (Ronneberger et al., 2015). However, segmentation requires labor-intensive pixel-level annotations and is more computationally demanding than object detection. For malaria diagnosis in thick smears, it is unclear whether precise pixel boundaries provide enough added value to justify this cost, especially given the small size and simple morphology of parasites.

## 2.5. Small Object Detection and P2 Heads

Many detection models struggle with very small objects (Zedda et al., 2023, 2025). Early ring forms of malaria parasites occupy only a small pixel area, and down-sampling through the feature hierarchy often removes the fine details needed for accurate detection (Zedda et al., 2023, 2025). Feature Pyramid Networks address multi-scale features, but deeper layers sacrifice spatial resolution. The P2 head is a solution found in modern architectures that adds a detection head at a higher resolution level of the feature pyramid (Mura et al., 2025). This preserves spatial detail before it is lost and improves sensitivity to tiny targets (Mura et al., 2025; Zedda et al., 2025). Evidence from computer vision supports this strategy, and some malaria studies have used variants of early-layer feature routing (Koirala et al., 2022). A formal validation of P2 heads in state-of-the-art malaria detectors is still missing, and this study provides that evaluation.

## 2.6. The Importance of External Validation and Domain Shift

Domain shift continues to be a major barrier to clinical deployment. Models often perform well on internal datasets but decline sharply when applied to data from different sites (Zedda et al., 2025; Sukumarran et al., 2024). Reported drops in performance can reach thirty percent across studies (Zedda et al., 2025; Okoronkwo, 2025; Sukumarran et al., 2024). Malaria microscopy is highly heterogeneous, with variation in staining, microscope models, camera sensors, image pipelines, slide preparation, and geographic differences in parasite appearance (Nakasi et al., 2025; Zedda et al., 2025; Okoronkwo, 2025). Many studies rely on single-dataset testing, which hides these issues and inflates expectations of real-world performance (Okoronkwo, 2025; Sukumarran et al., 2024; Zedda et al., 2025). Multi-center external validation is essential for assessing generalizability.

## 2.7. Computational Efficiency for Point of Care Deployment

Real-world deployment requires attention to computational constraints, especially in low-resource settings where malaria is most common (Lipsa et al., 2025). Many clinics operate without high-end hardware, and diagnostic systems must run on modest devices, including mobile phones or embedded processors (Alawfi, 2025; Koirala et al., 2022; Zedda et al., 2025). This creates a balance between accuracy, inference speed, memory use, and computational cost. Larger models may be accurate but impractical, while lightweight models

may be fast but insufficiently reliable. Research in malaria detection now regularly considers these trade-offs, and any thorough evaluation must include computational performance (Koirala et al., 2022).

### 2.8. Diagnosing Rare Species and Challenging Cases

A clinically useful system must detect rare species and difficult cases as reliably as common ones. Less prevalent species such as *P. malariae* and *P. ovale* are often misidentified even by skilled microscopists (Alawfi, 2025; Zedda et al., 2025). Early ring forms further increase difficulty. These challenges arise mainly from data imbalance, since most datasets overrepresent common species like *P. falciparum* (Ramarolahy et al., 2021). Models trained on imbalanced data tend to favor majority classes, which weakens performance on minority cases (Alawfi, 2025). Data augmentation can help by generating synthetic samples through rotation, scaling, or color variation (Islam et al., 2024). However, its effectiveness for rare Plasmodium species remains uncertain, which supports this study's focus on evaluating performance in rare and challenging scenarios.

### 2.9. Synthesis of Gaps in the Literature

Current research shows several gaps that limit progress toward clinically deployable malaria detection systems. Detection and segmentation methods have not been directly compared on thick smear images. Key features such as P2 heads remain insufficiently validated for small ring-stage parasites. Many studies omit external multi-center testing, which leads to an incomplete understanding of domain shift.Furthermore, high computational demands remain a barrier to deployment in low-resource settings. Performance on rare species and difficult cases is weakened by data imbalance. This study addresses these gaps to build a clearer evidence base for reliable and clinically useful diagnostic models.

## 3. Methods

### 3.1. Study Design and Validation Framework

We conducted a rigorous two-phase validation study to evaluate YOLOv12-based models for automated malaria parasite identification in thick blood smears. Our investigation addresses two key architectural questions: (1) whether instance segmentation provides advantages over standard object detection, and (2) whether incorporating high-resolution P2 detection heads improves identification of tiny parasites.

The validation framework consists of an internal phase using a Rwandan dataset spanning four *Plasmodium* species, followed by an external phase testing generalization across three geographically and institutionally distinct datasets. Critically, we employ a zero-shot evaluation protocol: models are trained exclusively on Rwandan data and evaluated on external datasets without any adaptation or fine-tuning, simulating real-world deployment where labeled data from the target institution may be unavailable.

## 3.2. Training and Validation Data

### 3.2.1. Rwandan Dataset Collection and Annotation

Our primary dataset originates from a collaboration with the Rwanda Biomedical Center (RBC), the national reference laboratory for malaria diagnostics. Blood samples were collected from patients presenting with fever at healthcare facilities, following established protocols documented in prior work by our lab (Akpo et al., 2024; Issah and Mukamakuza, 2026). Expert microscopists at RBC prepared Giemsa-stained thick blood smears after which we captured images at 100× magnification using an Olympus microscope.

Thick smears were specifically selected over thin smears due to their superior sensitivity for parasite detection, which is approximately 11-times higher (Poostchi et al., 2018), thus making them the preferred modality for clinical screening in endemic regions. The dataset includes 2,739 validated images with comprehensive annotations for four *Plasmodium* species: *P. falciparum* (Pf), *P. malariae* (Pm), *P. ovale* (Po), and *P. vivax* (Pv).

Image annotation was performed using the VGG Image Annotator (VIA) tool, with each parasite instance receiving both a bounding box (rectangular boundary) and a polygonal segmentation mask (precise contour). This dual annotation strategy enables training of both detection-only and segmentation-capable models. All annotations underwent iterative validation by RBC expert microscopists, with corrections incorporated until consensus was achieved. Of approximately 6,000 images collected to date, only the 2,739 fully validated images were utilized for model development, while the remaining images continue undergoing expert review for future work.

The dataset exhibits substantial class imbalance reflective of natural epidemiological patterns. *P. falciparum* is represented by 838 images containing 7,568 parasite instances, while *P. ovale* appears in 893 images with 2,353 instances. *P. malariae* comprises 834 images containing 1,802 instances, and *P. vivax* is severely underrepresented with only 174 images containing 669 instances (6.4% of data). This severe underrepresentation of *P. vivax* necessitated the species-targeted augmentation approach described in Section 3.4.2.

Data partitioning followed a 70-15-15 percentage split, yielding 1,915 training images, 410 validation images, and 411 test images.

### 3.2.2. External Validation Datasets

We assembled two publicly available thick smear datasets to evaluate model generalization across diverse acquisition conditions, geographic populations, and annotation protocols.

The Lacuna Malaria Dataset, compiled through the Lacuna Fund initiative, contains 3,925 thick smear images from Ghana (Lab, 2023; Nakasi et al., 2025), representing a distinct West African epidemiological context. Images were provided with pre-formatted YOLO bounding box annotations for only the *P. falciparum* and White Blood Cells (WBCs).

The FASTMAL Clinical Microscopy Dataset, developed by University College London (UCL) researchers, provides high-resolution (2560×2160) thick blood film images from *P. falciparum* infected patients with detailed rectangular bounding box annotations (Manescu et al., 2020b). Images were captured using extended depth-of-field microscopy with z-stack projection. To ensure annotation quality and format compatibility, we: (i) converted TIFF images to JPEG format, (ii) transformed proprietary JSON annotations to COCO format,

and (iii) filtered out all ambiguous annotations (crowd instances, background, and ignore regions), retaining only cleanly annotated parasite and white blood cell instances.

Table 1 summarizes the key characteristics of these external validation datasets, including species coverage, dataset size, geographic origin, and annotation methodology.

Table 1: External Validation Datasets Summary

| Dataset | Species Covered | Image Count | Source Region | Annotation Type |
|---------|-----------------|-------------|---------------|-----------------|
| Lacuna | *P. falciparum* | 3,925 | Ghana | Bounding Boxes |
| FASTMAL | *P. falciparum* | 243 | Nigeria | Bounding Boxes |

A significant limitation is the absence of publicly available thick smear datasets containing *P. malariae* or *P. ovale* with object-level annotations. We found publicly available datasets from the National Library of Medicine (NLM) containing *P. falciparum* and *P. vivax* (NLM), however persistent EXIF orientation metadata conflicts prevented reliable spatial alignment between images and annotations despite multiple preprocessing attempts. Consequently, external validation is restricted to *P. falciparum* only, with *P. vivax*, *P. malariae*, and *P. ovale* performance assessed only on the internal Rwandan test set.

### 3.3. Model Architectures and P2 Head Integration

#### 3.3.1. YOLOv12 Framework Selection

We selected YOLOv12n as our base architecture due to its recently introduced Attention-Centric design (Tian et al., 2025), which replaces traditional convolutional layers with Area Attention (A2) mechanisms. This architectural innovation reduces computational complexity while maintaining or improving detection accuracy, particularly relevant for deployment in resource-constrained clinical settings.

The standard YOLOv12 architecture employs a Feature Pyramid Network (FPN) with detection heads at three scales: P3 (stride 8), P4 (stride 16), and P5 (stride 32). These strides correspond to 1/8th, 1/16th, and 1/32nd of the input resolution respectively.

#### 3.3.2. Addressing the Small Object Challenge with P2 Heads

*Plasmodium* parasites, particularly *P. falciparum* ring-stage trophozoites, are exceptionally small objects in microscopy images. At 2048×2048 input resolution, early-stage rings frequently occupy fewer than 20×20 pixels. The standard P3-P5 detection pyramid operates at increasingly coarse spatial resolutions, potentially losing fine morphological details necessary to distinguish parasites from staining artifacts or cellular debris.

To preserve high-resolution spatial information, we augmented both detection and segmentation architectures with a P2 detection head operating at stride 4 (512×512 effective feature map resolution). This modification creates four model variants: YOLOv12-Obj-N (standard object detection with P3-P5 heads only), YOLOv12-Obj-N-P2 (detection with added P2 head), YOLOv12-Seg-N (instance segmentation with P3-P5 heads), and YOLOv12-Seg-N-P2 (segmentation with added P2 head).

### 3.4. Data Preprocessing and Training Protocol

#### 3.4.1. Image Standardization

All preprocessing operations were executed through Roboflow, which automatically adjusts bounding box and polygon annotations to match geometric transformations. Images were resized to 2048×2048 pixels using a letterbox method ("Fit with Black Edges" in Roboflow terminology). This approach scales the longer image dimension to 2048 pixels, proportionally scales the shorter dimension, then pads remaining space with zero-valued (black) pixels. Critically, this preserves aspect ratios and prevents morphological distortion of parasite structures that would occur with stretch-based or crop-based resizing methods.

Microscope cameras embed rotation metadata (EXIF tags) that can create inconsistent coordinate systems across images. We applied auto-orientation preprocessing to standardize pixel layouts, preventing systematic annotation misalignment (Roboflow, 2020).

#### 3.4.2. Addressing Class Imbalance Through Differential Augmentation

The extreme scarcity of *P. vivax* examples (121 training images versus 586-625 for other species) necessitated a targeted augmentation strategy. Rather than applying uniform augmentation factors, we implemented species-specific augmentation (Issah and Mukamakuza, 2026), with expansion factors proportional to underrepresentation: *P. falciparum*, *P. malariae*, and *P. ovale* received 3-fold expansion, while *P. vivax* received 10-fold expansion.

This applied augmentations are given in was Table 2 with values informed by augmentation protocols validated in prior malaria detection studies (Zedda et al., 2023). Roboflow generates augmented variants by randomly sampling from specified transformation ranges for each original training image, while validation and test sets remain unaugmented to ensure evaluation realism.

Table 2: Augmentation Transformation Specifications

| Augmentation | Values | Purpose |
| --- | --- | --- |
| Rotation | Discrete: 90°, 180°, 270° | Accounts for varying slide placement |
| Hue shift | Continuous: ±20° | Simulate stain and light variation |
| Saturation | Continuous: ±30% | Represents stain color variability |
| Brightness | Continuous: ±20% | Simulates variable illumination intensity |

These values were adopted from YOLO-PAM (Zedda et al., 2023), which demonstrated their effectiveness for Giemsa-stained microscopy. Importantly, we excluded shearing, perspective transforms, and other geometric distortions explicitly discouraged in that study, as they generate biologically implausible parasite morphologies. The final balanced and augmented dataset is given in Table 3.

#### 3.4.3. Model Training Configuration

All models were trained for 70 epochs on an NVIDIA H100 GPU (80GB VRAM) using consistent hyperparameters to ensure fair comparison. We employed Stochastic Gradient Descent (SGD) with momentum of 0.937 and weight decay of $5 \times 10^{-4}$. The batch size was

Table 3: Final Training Dataset Composition After Species-Specific Augmentation

| Species | Original | Expansion Factor | Final Train | Final Val | Final Test |
|---|---|---|---|---|---|
| *P. falciparum* | 586 | ×3 | 1,761 | 125 | 126 |
| *P. malariae* | 583 | ×3 | 1,752 | 125 | 125 |
| *P. ovale* | 625 | ×3 | 1,875 | 134 | 134 |
| *P. vivax* | 121 | ×10 | 1,210 | 26 | 26 |
| Totals | 1,915 | — | 6,598 | 410 | 411 |

set to 6, constrained by the 2048×2048 resolution, and mixed-precision training (FP16) was enabled to accelerate computation.

Initial experiments revealed training instability (exploding gradients manifesting as NaN losses) for P2-augmented models when using the standard learning rate of 0.01. This instability stems from the additional high-resolution detection head introducing steeper gradient magnitudes. To stabilize training without compromising convergence, we reduced the initial learning rate to 0.005 for P2 models only, while maintaining 0.01 for baseline models. This adjustment resolved gradient explosions and ensured all models converged within the 70-epoch window.

Training optimized a composite objective combining classification, localization, and (for segmentation models) mask prediction losses:

$$\mathcal{L}_{\text{total}} = \lambda_{\text{cls}}\mathcal{L}_{\text{cls}} + \lambda_{\text{box}}\mathcal{L}_{\text{box}} + \lambda_{\text{dfl}}\mathcal{L}_{\text{dfl}} + \lambda_{\text{mask}}\mathcal{L}_{\text{mask}} \tag{1}$$

The classification loss $\mathcal{L}_{\text{cls}}$ ($\lambda = 0.5$) employs multi-class cross-entropy for species classification. The bounding box loss $\mathcal{L}_{\text{box}}$ ($\lambda = 7.5$) uses Complete Intersection-over-Union (CIoU) for accurate localization regression. The distribution focal loss $\mathcal{L}_{\text{dfl}}$ ($\lambda = 1.5$) refines localization precision. For segmentation models, the mask loss $\mathcal{L}_{\text{mask}}$ applies binary cross-entropy for pixel-wise segmentation. Training progress was monitored via Weights & Biases for reproducibility.

### 3.5. Performance Metrics and Clinical Interpretation

We evaluated models using four complementary metrics, each addressing distinct clinical requirements.

#### 3.5.1. RECALL (SENSITIVITY)

Recall, defined as the ratio of true positives to the sum of true positives and false negatives, serves as our primary safety metric. In infectious disease screening, failing to identify an infected patient (false negative) can result in untreated infection and potential mortality. Therefore, models with high recall at the expense of modest precision are clinically preferable. For multi-species detection, species-specific recall indicates whether the model reliably identifies challenging species, and is computed as:

$$\text{Recall} = \frac{\text{True Positives}}{\text{True Positives} + \text{False Negatives}} \tag{2}$$

### 3.5.2. Mean Average Precision at IoU 0.5 (mAP@50)

This metric measures successful detection under a lenient spatial overlap criterion (50% Intersection-over-Union). For malaria screening, identifying the approximate parasite location is clinically sufficient, whereas exact pixel boundaries are less critical than flagging the infected cell for microscopist verification. Higher mAP@50 indicates robust parasite localization across diverse morphologies and imaging conditions, and is computed as:

$$\text{mAP@50} = \frac{1}{N_{\text{classes}}} \sum_{i=1}^{N_{\text{classes}}} \text{AP}_i(0.5) \tag{3}$$

### 3.5.3. Precision (Positive Predictive Value)

Precision quantifies the reliability of positive predictions. While important for minimizing false alarms, precision is secondary to recall. Missed parasites are more critical than a few misclassified artifacts. Precision is computed as:

$$\text{Precision} = \frac{\text{True Positives}}{\text{True Positives} + \text{False Positives}} \tag{4}$$

### 3.5.4. Mean Average Precision at IoU 0.5-0.95 (mAP@50-95)

This metric averages precision across stringent IoU thresholds (50%-95% overlap). For tiny objects like malaria parasites, achieving 95% IoU is mathematically challenging, given that single-pixel shifts dramatically reduce overlap. While standard in computer vision benchmarking, mAP@50-95 is less clinically meaningful than mAP@50 for object detection applications like this, where we are interested in counting. It is computed as:

$$\text{mAP@50-95} = \frac{1}{10} \sum_{t=0.5, 0.55, \ldots, 0.95} \text{AP}(t) \tag{5}$$

### 3.6. Reproducibility Statement

Implementation utilized PyTorch with Ultralytics YOLOv12. All hyperparameters, and training configurations are fully specified above.

## 4. Results and Discussion

### 4.1. Internal Validation (Rwandan Test Set)

#### 4.1.1. Baseline Model Comparison (Detection vs Segmentation)

Table 4 presents overall performance of baseline detection (YOLOv12-Obj-N) and segmentation (YOLOv12-Seg-N) models.

Aggregate metrics show minimal differences, but per-species analysis (Table 5) reveals important patterns.

Segmentation substantially benefits rare species: *P. malariae* (+7.9%) and *P. falciparum* (+2.5%). Pixel-level supervision helps distinguish rare parasites from complex backgrounds. Conversely, segmentation reduces *P. vivax* performance (0.940 to 0.821), likely because its

Table 4: Baseline Model Performance Comparison (Box Metrics, All Classes)

| Model | Architecture | Precision | Recall | mAP@50 | mAP@50-95 |
|---|---|---|---|---|---|
| YOLOv12-Obj-N | Detection | **0.837** | 0.820 | **0.874** | 0.636 |
| YOLOv12-Seg-N | Segmentation | 0.800 | **0.829** | 0.859 | **0.640** |

Table 5: Baseline Model Performance by Species (Box mAP@50)

| Species | Obj-N | Seg-N |
|---|---|---|
| *P. falciparum* | 0.752 | **0.771** |
| *P. malariae* | 0.837 | **0.903** |
| *P. ovale* | **0.921** | 0.897 |
| *P. vivax* | **0.940** | 0.821 |

distinctive amoeboid morphology is well-captured by bounding boxes. **RQ1:** Segmentation provides selective benefits for rare species but not universally. It is most useful when morphological precision is required.

### 4.1.2. P2 HEAD IMPACT ON SEGMENTATION MODELS

Table 6 compares segmentation models with and without P2 heads.

Table 6: P2 Head Impact on Segmentation Models (All Classes)

| Metric | Seg-N | Seg-N-P2 | Δ |
|---|---|---|---|
| Box mAP@50 | 0.859 | **0.888** | **+2.9%** |
| Box mAP@50-95 | 0.640 | **0.676** | **+3.6%** |
| Mask mAP@50 | 0.859 | **0.888** | **+2.9%** |
| Mask mAP@50-95 | 0.605 | **0.656** | **+5.1%** |

The P2 head consistently improves all metrics, with mask improvements (+5.1%) exceeding box improvements (+3.6%), suggesting benefits primarily for boundary precision.

### 4.1.3. PER-CLASS PERFORMANCE WITH P2 HEAD ON SEGMENTATION MODELS

Table 7 reveals species-specific heterogeneity.

*P. vivax* shows dramatic improvement (+10.9% box, +16.8% mask), an indication that high-resolution P2 features enable better generalization particularly for Pv's complex amoeboid morphology. Other species show modest box improvements (0.4–0.8%) but substantial mask improvements (3.1–3.3%), indicating P2 primarily refines boundary precision. For ring-stage *P. falciparum*, the 3.3% mask improvement represents meaningful morphological refinement.

Table 7: Per-Class P2 Impact on Segmentation Models

| Class | Metric | Seg-N | Seg-N-P2 | Δ |
|---|---|---|---|---|
| *P. falciparum* | Box mAP@50 | 0.771 | **0.779** | **+0.8%** |
| | Mask mAP@50-95 | 0.444 | **0.477** | **+3.3%** |
| *P. malariae* | Box mAP@50 | 0.903 | **0.907** | **+0.4%** |
| | Mask mAP@50-95 | 0.629 | **0.660** | **+3.1%** |
| *P. vivax* | Box mAP@50 | 0.821 | **0.930** | **+10.9%** |
| | Mask mAP@50-95 | 0.501 | **0.669** | **+16.8%** |

### 4.1.4. P2 Head Impact on Detection Models

Table 8 shows more complex effects for detection-only models.

Table 8: P2 Head Impact on Detection Models (All Classes)

| Metric | Obj-N | Obj-N-P2 | Δ |
|---|---|---|---|
| Precision | **0.837** | 0.787 | −5.0% |
| Recall | 0.820 | **0.843** | **+2.3%** |
| mAP@50 | **0.874** | 0.862 | −1.2% |
| mAP@50-95 | **0.636** | 0.615 | −2.1% |

P2 increases recall by 2.3% but reduces precision by 5.0%, decreasing overall mAP. This recall-precision trade-off indicates that for detection-only models, higher sensitivity is outweighed by false positives.

### 4.1.5. Computational Cost

Table 9 quantifies P2 overhead.

Table 9: Computational Trade-offs of P2 Integration

| Metric | Seg-N | Seg-N-P2 | Change |
|---|---|---|---|
| Inference Time (ms) | 25.4 | 30.1 | +18.5% |
| Parameters (M) | 2.76 | 2.83 | +2.5% |
| GFLOPs | 7.9 | 8.4 | +6.3% |

P2 adds modest parameter overhead (2.5%) but 18.5% latency increase. This suggests burden stems from higher-resolution feature maps rather than parameters. The proposed YOLOv12-Seg-N-P2 model maintains a compact footprint with only 2.83 million parameters and 8.4 GFLOPs, significantly more efficient than standard medical detection architectures like Faster R-CNN which has over 40M parameters, or U-Net ensembles.

While inference latency on GPU ($\approx$30ms) indicates high feasibility for real-time deployment in well-equipped microscopy labs, deployment to low-resource settings requires additional considerations. Modern mobile Neural Processing Units (NPUs) typically support

>10 Tera Operations Per Second (TOPS), suggesting this model is theoretically capable of real-time inference on edge devices like smartphones through the standard ONNX or TFLite optimization pipelines. However, empirical validation on target hardware like mobile phones was not conducted in this study as it is the next phase of our work, remains necessary for deployment in resource-constrained point-of-care settings.

**RQ4:** Computational overhead is modest and acceptable for deployment in well-equipped microscopy labs. Theoretical analysis suggests feasibility for mobile/edge deployment, but empirical validation on low-resource devices is required before point-of-care deployment.

### 4.2. External Validation (Lacuna & FASTMAL)

External validation used detection models only (external datasets lack masks). Zero-shot transfer (no fine-tuning) assesses domain shift robustness.

#### 4.2.1. PERFORMANCE ON LACUNA AND FASTMAL

Table 10 presents results for *P. falciparum* and WBC detections.

Table 10: External Validation Results (Zero-Shot Transfer)

| Dataset | Model | Class | Precision | Recall | mAP@50 |
|---|---|---|---|---|---|
| Lacuna | Obj-N | *P. falciparum* | **0.140** | **0.174** | **0.0998** |
| | | WBC | 0.341 | 0.441 | 0.366 |
| | Obj-N-P2 | *P. falciparum* | 0.117 | 0.0919 | 0.0690 |
| | | WBC | 0.425 | 0.379 | 0.369 |
| FASTMAL | Obj-N | *P. falciparum* | **0.169** | **0.323** | **0.128** |
| | | WBC | 0.697 | 0.858 | **0.778** |
| | Obj-N-P2 | *P. falciparum* | 0.138 | 0.252 | 0.0975 |
| | | WBC | **0.771** | 0.856 | **0.806** |

*P. falciparum* performance collapses on external datasets: mAP@50 drops to 0.0998 (Lacuna) (Lab, 2023) and 0.128 (FASTMAL). P2 models underperform baselines by 31–44%. In contrast, WBC detection generalizes substantially better: FASTMAL WBC mAP reaches 0.778–0.806 (comparable to internal 0.921). Lacuna WBC mAP of 0.366–0.369 is lower but respectable. This asymmetry reflects object size and morphological complexity: larger, simpler WBCs transfer better than tiny, staining-dependent parasites.

#### 4.2.2. DOMAIN SHIFT QUANTIFICATION

Table 11 quantifies the severity.

Table 11: Domain Shift Magnitude Across External Datasets

| Dataset | Pf mAP@50 (Internal) | Pf mAP@50 (External) | Relative Drop |
|---|---|---|---|
| Lacuna | 0.75–0.78 | ∼0.10 | ∼87% |
| FASTMAL | 0.75–0.78 | ∼0.13 | ∼83% |

Both datasets exhibit catastrophic Pf degradation (83–87% drop), reflecting fundamental distribution shifts from differences in imaging equipment, staining, resolution, and presentation. The baseline (Obj-N) consistently outperforms P2 variants, suggesting P2's high-resolution features overfit to Rwandan characteristics (staining intensity, camera noise, lighting) and fail to generalize.

To investigate the mechanism of this failure, we conducted a detailed visual analysis of the dataset distributions (Figure 1). The internal Rwandan training data (Figure 1A) consists of standard rectangular fields captured via a high quality camera-mounted Olympus microscope with a characteristic organic pinkish/purple color. In contrast, the external datasets exhibit severe hardware-driven shifts. Lacuna images (Figure 1B) were captured via mobile phones (Samsung S8+, Redmi) held to the eyepiece of an Olympus CX-23 microscope (Lab, 2023), introducing circular vignetting, lens distortion, and uneven illumination absent in our training data. Conversely, FASTMAL images (Figure 1C) were acquired using a high-end Olympus BX63 with a PCO Edge 5.5c camera and processed using Wavelet Extended Depth of Field (EDoF) (Manescu et al., 2020b). This creates a "grainy" texture and dark blue/gray hue. These differences in pixel scaling, geometry, and texture explain the drop in detection performance and poor generalization without domain adaptation, especially for the P2 head which relies on fine-grained morphological features.

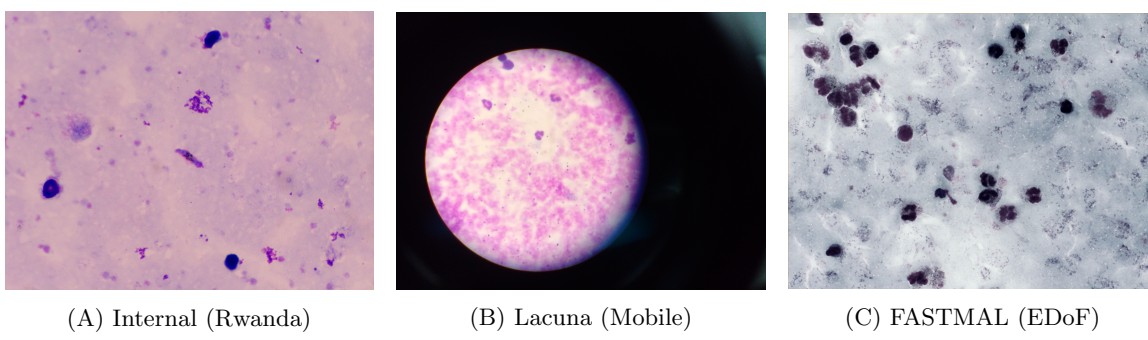

(A) Internal (Rwanda)          (B) Lacuna (Mobile)          (C) FASTMAL (EDoF)

Figure 1: Visual analysis of domain shifts driving performance degradation. (A) Internal Rwanda dataset (Standard microscopy, rectangular field). (B) Lacuna dataset (Mobile phone capture) (Lab, 2023), exhibiting circular vignetting and geometric distortion. (C) FASTMAL dataset (Extended Depth of Field), exhibiting synthetic "grainy" texture and dark blueish hue.

**RQ2:** P2 is effective for segmentation on source data, particularly for rare species (+10.9% Pv) and boundary refinement (+3.3% Pf mask). However, P2 fails on external data (31–44% worse), indicating overfitting to Rwandan acquisition characteristics.

**RQ3:** Zero-shot transfer shows severe degradation for parasites (83–87%) but lower degradation for WBCs ( 25–60%), underscoring tiny objects' vulnerability to domain shift. Current models require fine-tuning or domain adaptation before multi-site deployment.

### 4.2.3. Domain Adaptation via Stain Normalization and Rescaling

The severe domain shift identified above motivates two preprocessing interventions, both clinically practical because WBCs are always present (with 20-40% lymphocytes):

(1) **WBC-anchored stain normalization**, which transfers Rwanda's colorimetric statistics onto the external test images using white blood cell (WBC) bounding-box crops to compute stain statistics; three methods were evaluated: Macenko (Macenko et al., 2009), Reinhard (Reinhard et al., 2001), and Vahadane (Vahadane et al., 2016);

(2) **Lymphocyte-based rescaling**, which uses the mean WBC lymphocyte diameter in the Rwanda training set (105.67 px) as a biological scale reference to rescale external images to match internal pixel resolution. The rescaling factors for FASTMAL and Lacuna were determined as 0.830 and 0.709 respectively. Because Macenko and Vahadane apply OD-space statistics transfer with fixed Giemsa stain vectors, they produce identical outputs for thick Giemsa smears and are reported together.

Table 12 presents mAP@50 test results for all preprocessing conditions applied to the original object detection zero-shot baseline model (Obj-N).

Table 12: Effect of stain normalization and lymphocyte-based rescaling on zero-shot transfer (Obj-N). Macenko and Vahadane produce identical outputs for Giemsa thick smears and are shown as a single entry. Baseline values reproduced from Table 10.

| Dataset | Preprocessing | Pf mAP@50 | WBC mAP@50 |
|---------|---------------|-----------|------------|
| FASTMAL | Baseline (none) | 0.128 | 0.778 |
|         | Rescaling only | 0.127 | 0.778 |
|         | Macenko / Vahadane | **0.200** | **0.808** |
|         | Reinhard | 0.139 | 0.804 |
|         | Rescaling + Macenko / Vahadane | **0.201** | 0.805 |
|         | Rescaling + Reinhard | 0.140 | 0.808 |
| Lacuna  | Baseline (none) | 0.100 | 0.366 |
|         | Rescaling only | 0.097 | 0.367 |
|         | Macenko / Vahadane | 0.095 | **0.532** |
|         | Reinhard | 0.096 | 0.411 |
|         | Rescaling + Macenko / Vahadane | 0.096 | 0.530 |
|         | Rescaling + Reinhard | 0.096 | 0.415 |

Stain normalization substantially improves WBC detection on Lacuna and modestly on FASTMAL. For *P. falciparum*, Macenko/Vahadane normalization yields a 50% gain on FASTMAL, yet Pf mAP@50 remains critically low ($\leq 0.201$). On Lacuna, no preprocessing condition improves Pf detection beyond baseline. Lymphocyte-based rescaling alone has no measurable effect on either class, confirming that YOLO's multi-scale training confers sufficient scale invariance. These findings indicate that although stain shift is a secondary contributor to Pf failure, the primary obstacle is the structural domain gap that these preprocessing steps alone cannot overcome.

### 4.3. Limitations and Future Directions

While external validation was restricted to *P. falciparum* due to data availability, this focus is clinically and technically justified. *P. falciparum* accounts for the vast majority of malaria mortality (World Health Organization, 2024) and represents the most challenging computer vision task due to the minute size of early ring stages ($<2~\mu$m) and their resemblance to staining artifacts (Delahunt et al., 2024). Distinguishing these tiny targets from background noise is significantly harder than identifying the larger, distinct morphologies of *P. vivax* or *P. malariae*. Thus, performance on *P. falciparum* serves as the primary stress-test for model robustness.

Our visual analysis (Figure 1) suggested that performance degradation is likely driven by geometric (pixel scaling) and colorimetric (stain) shifts. To bridge this gap, we implemented lymphocyte-based rescaling, and three stain normalization methods to align external datasets with the internal training distribution. As shown in Section 4.2.3, stain normalization substantially recovers WBC detection, and improves but does not resolve the catastrophic Pf degradation, particularly on the Lacuna mobile-phone dataset.

### 4.4. Clinical and Validation Study Implications

On the source domain, YOLOv12-Seg-N-P2 achieves strong performance (mAP@50 0.888), especially for challenging cases like Pv and Pf ring forms. However, external validation reveals these models are not deployment-ready without further validation. The severe domain shift reflects a fundamental medical AI challenge: models learn dataset-specific characteristics rather than generalizable patterns. This underscores the critical importance of external validation. For clinical adoption, models must be validated across institutions, deployed with on-site fine-tuning, or used as decision-support tools with human expertise. Our findings provide evidence-based guidance: segmentation models are preferable when species precision is required, P2 heads improve source performance but reduce generalizability, and current models require domain adaptation before multi-site deployment.

## 5. Conclusion

This study provides a systematic validation of modern deep learning models for automated malaria microscopy and addresses several long-standing gaps in the field. By evaluating detection and segmentation approaches on thick smear images, the work clarifies the practical differences between bounding boxes and pixel-level masks for parasite localization. The analysis of P2 heads within the YOLOv12 architecture shows how early feature maps can improve detection of small ring-stage parasites. The cross-dataset evaluation demonstrates the extent of domain shift and highlights the need for multi-center validation before clinical deployment. The study also shows that computational efficiency remains a key constraint for point-of-care use and that performance on rare species and low-parasitemia slides continues to be a major challenge. Together, these findings form an evidence base that can guide future research and support the development of diagnostic systems that perform reliably across diverse clinical settings.

## Acknowledgments

We thank the Rwanda Biomedical Center for their collaboration and data provision. We acknowledge support from Carnegie Mellon University Africa. We are especially grateful to Professor Charles B. Delahunt for his steady guidance, technical reviews, and constructive feedback throughtout this work

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
