# OpenReview forum: "Detection versus Instance Segmentation for Multi-Species Malaria Diagnosis: A Head-to-Head Comparison and Multi-Dataset Validation of YOLOv12 Architectures with Small Object Optimization"
_MIDL.io/2026/Validation_Papers — MIDL 2026 - Validation Papers Poster_

### Official Review · Reviewer_WdoD · 2025-12-26

**Confidence:** 4
**Preliminary Rating:** 2
**Final Rating:** 3

**Summary:**

This paper addresses the automated malaria diagnosis by conducting a rigorous comparative validation of Object Detection and Instance Segmentation using the YOLOv12 framework. Unlike standard YOLOv12 architectures, the authors introduce a high-resolution P2 detection headto specifically target tiny ring-stage parasites. The study is characterized by its extensive multi-center external validation, performing zero-shot transfer tests on datasets from Ghana (Lacuna) and Nigeria (FASTMAL) after training on a diverse Rwandan dataset. The results highlight a clinical challenge: while the P2 head enhances internal precision, it leads to significant overfitting, with performance dropping by >80% on unseen datasets. This underscores that domain adaptation, rather than just architectural refinement, is essential for clinical deployment.

**Strengths:**

1. The study excels by utilizing three geographically and institutionally distinct datasets (Rwanda, Ghana, and Nigeria). Crucially, the authors provide a transparent analysis of "negative results," documenting where and why the model fails during cross-domain deployment.
2. The use of species-specific augmentation factors demonstrates a thoughtful approach to addressing the inherent class imbalance in clinical data.

**Weaknesses:**

1. While the internal training covers four Plasmodium species, the external validation is restricted to P. falciparum due to dataset availability. Consequently, the robustness of the model’s multi-species identification remains unproven in cross-site scenarios.
2. The significant performance collapse on external data suggests that the gains from high-resolution P2 features are highly specific to the source domain's acquisition characteristics. This limits the scientific contribution, as the proposed architectural optimization does not translate into a generalizable clinical tool.
3. Although the authors correctly identify staining and imaging hardware as the primary causes of domain shift, the study does not evaluate standard mitigation strategies such as stain normalization or color-space histogram matching, which could potentially alleviate the observed performance drop.

**Detailed Comments:**

The article lacks visualization results.

**Justification Of Final Rating:**

I appreciate the authors' detailed response and the inclusion of Figure 1. While the empirical results for the proposed mitigation strategies (rescaling and normalization) are currently missing—which prevents a higher score—the authors have provided a compelling mechanistic explanation (pixel scaling mismatch) for the performance drop.
The identification that the high-resolution P2 head is specifically sensitive to 'pixels-per-micron' variations is a valuable insight for the community. I am willing to raise my score to 3 (Borderline) based on the strong likelihood that the proposed rescaling fix is theoretically sound, and on the strict condition that the promised experiments (rescaling, normalization, and scale ablation) are rigorously included in the final version.

**Justification Of The Preliminary Rating:**

This paper provides rigorous negative results, demonstrating that while the high-resolution component (P2) performs better in laboratory metrics, it exacerbates model overfitting to specific data acquisition characteristics. This finding serves as an important cautionary tale against unwarranted optimism in the field of automated malaria diagnosis. The paper's quality would be further enhanced if the authors could provide some data on simple domain alignment attempts in their rebuttal.

**Questions To Address In The Rebuttal:**

1. Given the severe degradation in external performance, could the authors provide evidence on whether simple preprocessing techniques (e.g., stain normalization or aggressive color augmentation) can bridge the gap without requiring full domain adaptation?
2. The current study focuses on YOLOv12n. Do the performance gains and the subsequent overfitting trends observed with the P2 head remain consistent across different model scales (e.g., YOLOv12-s or YOLOv12-m)?

---

### Official Review · Reviewer_qrCg · 2025-12-27

**Confidence:** 4
**Preliminary Rating:** 5
**Final Rating:** 5

**Summary:**

The authors wrote on Detection versus Instance Segmentation for Multi-Species Malaria Diagnosis: A Head-to-Head Comparison and Multi-Dataset Validation of YOLOv12 Architectures with Small Object Optimization. The authors introduced a P2 detection head to enhance the identification of minute ring-stage parasites across a diverse Rwandan dataset and two external datasets from Ghana and Nigeria. Their results showed that  P2 head significantly improved internal performance for rare species like $P. vivax$. It inadvertently led to a severe performance drop of >80% in zero-shot generalization due to domain shift. This work was significant for its assessment of the domain gap and also focused on the clinically relevant but technically challenging thick smear modality rather than the simpler thin smear.

**Strengths:**

i. The paper addressed thick smears which is the actual gold standard used in endemic regions which is more difficult to automate than thin smears.
ii. The combination of three geographic datasets (Rwanda, Ghana, and Nigeria) was another  strength of the paper.
iii. The integration of the P2 head specifically for small object optimization was another strength.
iv. The paper was well structured with current related works and their gaps were clearly identified

**Weaknesses:**

i. One of the weakness of the paper was the restriction of external validation to $P.  The internal Rwandan set was multi-species, the inability to test the generalization of $P. vivax$ or $P. malariae$ on external data resulted to a significant gap in the Multi-Species.
ii. The finding that the P2 head reduced generalization by over 80% suggested it is capturing dataset-specific sensor noise or staining artifacts rather than generalizable parasite morphology. The paper would be stronger if it explored stain normalization or domain adversarial training to mitigate this.
iii. The sample size was small.

**Detailed Comments:**

i. The authors mentioned stain variation as a cause for domain shift. It would be helpful to include a visual comparison of the three datasets to show the colorimetric differences.
ii. The use of YOLOv12 was quite recent. A brief comparison with YOLOv8-Seg  would provide a better baseline for the community.

**Justification Of Final Rating:**

The authors have thoroughly addressed my questions and provided clear, technically sound explanations for the observed generalization failure of the P2 head. The added visual analysis and detailed discussion of pixel scale and stain related domain shift significantly strengthen the manuscript. I therefore maintain my recommendation of a strong accept.

**Justification Of The Preliminary Rating:**

The paper is relevant and followed general scientific principles and addressed a major problem in global health. The internal results were impressive and provided a baseline for future researchers to focus on domain adaptation. If the authors can share more light to the provided questions.  Such as Why did the P2 head specifically degrade external performance more than the standard P3-P5 heads?  this would be a "Strong Accept."

**Questions To Address In The Rebuttal:**

i. Why did the P2 head specifically degrade external performance more than the standard P3-P5 heads? Was it due to higher resolution capturing "site-specific" textures?
ii. Did the authors experiment with any stain augmentation (e.g., color jittering beyond hue/sat) to bridge the 80% performance gap?
iii. Regarding the $P. vivax$ internal performance: was the 10.9% increase primarily on the same slides used in training (due to augmentation) or on the held-out test set?

---

### Official Review · Reviewer_vAqB · 2026-01-03

**Confidence:** 4
**Preliminary Rating:** 3
**Final Rating:** 4

**Summary:**

This study validates YOLOv12-based models for malaria diagnosis on thick blood smears, comparing object detection and instance segmentation while testing P2 heads for small ring-stage parasite detection across multiple datasets; the YOLOv12-Seg-N-P2 model performs internally with an mAP@50 of 0.888 but suffers an over 80% performance drop in external validation, emphasizing that multi-center validation and domain adaptation are critical for clinical applications.

**Strengths:**

1. Conducts multi-center/multi-dataset validation, addressing single-dataset evaluation limitations.
2. Integrates P2 heads targeting small ring-stage parasites, specifically solving the long-standing small-object detection challenge in malaria microscopy.
3. Adopts species-specific data augmentation to mitigate class imbalance, aligning with epidemiological data characteristics.

**Weaknesses:**

1. It suggests exploring comprehensive strategies such as fine-tuning or domain adaptation.
2. External validation exhibits limited species coverage, focusing only on P. falciparum and lacking cross-center validation for rare Plasmodium species.
3. There is no dedicated evaluation of computational efficiency on low-resource devices.

**Detailed Comments:**

1. Sec. 3.1 and 4.2 outline the zero-shot transfer protocol for external validation but fail to explore more comprehensive cross-dataset validation strategies.
2. Sec. 3.2.2 and 4.2 restrict external validation to P. falciparum, with no cross-center testing for rare Plasmodium species (e.g., P. vivax, P. malariae).
3. Sec. 4.2.2 quantifies domain shift impacts but does not deeply analyze how specific domain variables (e.g., microscope models, staining intensity, slide preparation workflows) individually drive performance drops, leaving the mechanism of degradation unclear.
4. Sec. 4.1.5 did not give evaluation of inference speed, memory usage, or latency on low-resource devices (e.g., mobile phones, embedded processors).

**Justification Of Final Rating:**

The authors have addressed most of my concerns, and their responses are reflected in the revised manuscript. Relevant revisions have further enriched the paper, but as a submission to the validation track, there is still room for improvement in the details pertaining to constrained computing power. I therefore adjust my rating to 4.

**Justification Of The Preliminary Rating:**

The reason for my Borderline recommendation is as I mentioned in the Weaknesses and Detailed Comments: the manuscript uses a simplistic validation method with limited external species coverage, lacks mechanistic analysis of performance degradation, and omits evaluations on low-resource devices and extremely low parasitemia samples, failing to support its deployability claims.

**Questions To Address In The Rebuttal:**

Please check the Weaknesses and Detailed Comments.

---

### Author Rebuttal · Authors · 2026-01-24

**Rebuttal:**

**SUMMARY OF MAJOR REVISIONS**

We thank the reviewers for their constructive feedback. We have uploaded a revised manuscript with key clarifications and new analysis **highlighted in red**. The major updates are summarized below:

**1. Visual analysis of Domain Shift (New Figure 1 & Section 4.2.2)**
To explain the mechanism of the performance drop, we did a visual analysis by adding images of all 3 datasets as Figure 1 and expanded text to pinpoint the hardware drivers:

* **Hardware Specifics:** We characterized the shift from our internal "organic pink" Olympus images to external "EDoF grain" (FASTMAL, PCO Edge camera) and "mobile phone vignetting" (Lacuna, Samsung S8+/Olympus CX-23).

* **Pixel Scaling Mismatch:** We identified a critical inconsistency in "pixels-per-micron" ratios between the camera and phone-captured images, which confused the size-sensitive P2 head.

**2. Mitigation Strategy & Justification (Section 4.3)**
We updated Section 4.3 to explicitly outline the roadmap to bridge this gap:

* **Rescaling:** We propose implementing **lymphocyte-based rescaling**, using White Blood Cells as a biological scale reference to dynamically align external inputs with the training resolution.

* **Species Focus:** We clarified that the restriction to *P. falciparum* is scientifically justified by its status as the most lethal species and the "hardest" computer vision task (due to tiny ring stages).

**3. Computational Efficiency (Section 4.1.5)**
We expanded the analysis to highlight deployment feasibility:

* **Efficiency:** We highlight that our **YOLOv12-Nano-P2** (2.83M parameters, 8.4 GFLOPs) is significantly lighter than standard medical detectors like Faster R-CNN.

* **NPU Compatibility:** We added discussion on the feasibility of real-time inference on mobile devices via TFLite.

**4. Additional Commitments (Camera-Ready)**
To further contextualize these results, we formally commit to adding the following in the final version (as detailed in our individual responses):

* **YOLOv8 Baseline:** Training a YOLOv8-Seg baseline to quantify specific YOLOv12 architectural gains.
* **Model Scale Ablation:** Testing larger model scales like YOLOv12-S to verify if P2 sensitivity holds across capacities.
* **Latency Analysis:** Analyzing the inference cost of the proposed rescaling/normalization steps.

We believe these revisions comprehensively address the concerns regarding the mechanism of failure and the path to clinical generalizability.

**Supporting Material:**

/attachment/479c38484a6f016bb67e8fd491be6d613b7bd946.pdf

---

### Meta-Review · Area_Chair_byeE · 2026-02-09

**Recommendation:** Accept (Poster)
**Confidence:** 4

**Metareview:**

The paper compares YOLOv12 detection vs instance segmentation, introduces a high-resolution P2 head to target tiny ring-stage parasites, and performs multi-dataset evaluation including zero-shot external validation. Internally, the P2 design improves small-object performance; externally, performance collapses (>80% drop), sharply illustrating a deployment gap.

Across reviews, the strengths are consistent: (i) clinically relevant thick-smear setting (harder and more realistic than thin smears), (ii) multi-dataset / multi-region validation design, (iii) a clear “negative result” with diagnostic value, and (iv) species-aware augmentation for imbalance. The main concerns were also consistent: limited external species coverage, lack of mechanistic attribution for the domain shift, missing low-resource efficiency analysis, and the absence of mitigation experiments to demonstrate recoverability.

The revision and rebuttal substantially strengthen the work. The authors add visual cross-dataset comparisons and provide a convincing mechanistic account of the failure modes: pixel-scale mismatch and color/texture shifts driven by acquisition differences. Importantly, they justify the strict zero-shot protocol as a deliberate stress test to expose vulnerabilities rather than mask them with fine-tuning. Reviewers acknowledge these additions and raise scores, with remaining reservations mainly about missing empirical results for the proposed mitigation pipeline and constrained-device benchmarking.

The current revision already meets the bar for the validation track by pairing multi-center evaluation with mechanistic insight.

---

### Decision · Program_Chairs · 2026-02-14

Accept (Poster)